

# The effects of a 4-week mesocycle of barbell back squat or barbell hip thrust strength training upon isolated lumbar extension strength

Alexander Hammond[1], Craig Perrin[1], James Steele[1], Jürgen Giessing[2], Paulo Gentil[3] and James P. Fisher[1]

[1] School of Sport, Health and Social Sciences, Southampton Solent University, Southampton, UK
[2] Institute of Sport Science, Universität Koblenz-Landau, Landau, Germany
[3] Faculty of Physical Education and Dance, Universidade Federal de Goiás, Goiania, Brazil

Corresponding author
James P. Fisher,
james.fisher@solent.ac.uk

## ABSTRACT

**Objectives:** Common exercises such as the barbell back squat (BBS) and barbell hip thrust (BHT) are perceived to provide a training stimulus to the lumbar extensors. However, to date there have been no empirical studies considering changes in lumbar extension strength as a result of BBS or BHT resistance training (RT) interventions.

**Purpose:** To consider the effects of BBS and BHT RT programmes upon isolated lumbar extension (ILEX) strength.

**Methods:** Trained male subjects ($n = 14$; $22.07 \pm 0.62$ years; $179.31 \pm 6.96$ cm; $79.77 \pm 13.81$ kg) were randomised in to either BBS ($n = 7$) or BHT ($n = 7$) groups and performed two training sessions per week during a 4-week mesocycle using 80% of their 1RM. All subjects were tested pre- and post-intervention for BBS and BHT 1RM as well as isometric ILEX strength.

**Results:** Analyses revealed that both BBS and BHT groups significantly improved both their BBS and BHT 1RM, suggesting a degree of transferability. However, the BBS group improved their BBS 1RM to a greater degree than the BHT group ($p = 0.050$; ~11.8 kg/10.2% vs. ~8.6 kg/7.7%, respectively). And the BHT group improved their BHT 1RM to a greater degree than the BBS group ($p = 0.034$; ~27.5 kg/24.8% vs. ~20.3 kg/13.3%, respectively). Neither BBS nor BHT groups significantly improved their isometric ILEX strength.

**Conclusions:** The present study supports the concept of specificity, particularly in relation to the movement mechanics between trunk extension (including pelvic rotation) and ILEX. Our data suggest that strength coaches, personal trainers, and trainees can self-select multi-joint lower-body trunk extension exercises based on preference or variety. However, evidence suggests that neither the BBS nor BHT exercises can meaningfully increase ILEX strength. Since strengthening these muscles might enhance physical and sporting performance we encourage strength coaches and personal trainers to prescribe ILEX exercise.

## INTRODUCTION

Low-back strength, particularly as a component part of core strength and stability, retains importance in athletic performance and thus strength and conditioning within sports (*Hibbs et al., 2008*). Indeed, the strength and cross-sectional area of the erector spinae and quadratus lumborum have been suggested to share 50% of the variance in sprint speed over 20 m (*Kubo et al., 2011*), a substantial contribution for an inconspicuous muscle during sprinting. Furthermore, elevated forces through the lumbar spine whilst blocking in American football, as well as during a golf swing, suggest that improving lumbar strength might be beneficial towards both enhancing performance and reducing risk of injury (*Gatt et al., 1997*; *Gluck, Bendo & Spivak, 2008*). Finally, simulation research suggests erector spinae weakness results in compensation from synergistic muscles, potentially causing an earlier onset of fatigue and exacerbating low-back pain and performance decrements (*Raabe & Chaudhari, 2018*).

Research has supported that common exercises such as the barbell back squat (BBS) place considerable stress on the lumbar musculature (*Cholewicki, McGill & Norman, 1991*) supported by the high levels of activation of the lumbar muscles when measured by electromyography (EMG; *Hamlyn, Behm & Young, 2007*; *Yavuz et al., 2015*). Indeed, as it is thought the lumbar extensor musculature is heavily involved in the BBS, many strength and conditioning coaches advocate the use of the BBS and deadlift exercises with a view to provide a training stimulus and increase the strength of the lumbar muscles (*Mayer, Mooney & Dagenais, 2008*).

Nevertheless, whilst the BBS exercise appears to be a hypothetical solution to strengthening the lumbar muscles (although no empirical evidence exists), alternative and perhaps superior exercises may be available. An exercise growing in popularity is the barbell hip thrust (BHT), which has been shown to produce greater EMG amplitude in the hamstring and gluteal muscles when compared to the BBS exercise (*Contreras et al., 2015*). The BHT has also been shown to be associated with performance markers such as acceleration ($r = 0.93$; *Loturco et al., 2018*). Furthermore, *Contreras et al. (2017)* recently compared 6-weeks of strength training using either a front squat or BHT exercise on performance markers in adolescent males. The authors reported favourable effect sizes (ES) for the BHT for 10 m and 20 m sprint times, and isometric mid-thigh pull, whereas the front squat exercise produced favourable ES for the vertical jump and front squat 3RM. *Andersen et al. (2018)* compared EMG activity during the barbell deadlift, hex bar deadlift, and BHT exercises. Although there were no statistically significant differences between the three exercises for erector spinae muscle activation, the BHT did elicit the greatest muscle activation during the concentric phase of the movement when the hip angle was approaching or exceeding 180 degrees. The authors suggest that the BHT likely promotes increased muscle activation in the upper phase due to the increased hip torque requirement in the end range of this horizontally loaded exercise. In contrast, during a deadlift or BBS exercise the lumbo-pelvic complex reaches complete hip extension in a vertical anatomical position meaning the forces produced by the barbell load the skeletal system and likely produce lower muscular activation.

However, the BBS and BHT both involve the lumbo-pelvic complex in a compound movement integrating both hip and lumbar extension (i.e. trunk extension). Previous research considering both resistance machines (*Graves et al., 1994*) and free-weight exercise (the Romanian deadlift; *Fisher, Bruce-low & Smith, 2013*) has suggested that trunk extension exercise (hip and lumbar extension) that permits rotation of the pelvis is not efficacious in increasing isolated lumbar extension (ILEX) strength. This is likely a result of hamstring and gluteal contribution via pelvic rotation, rather than ILEX. Indeed, research has supported that there is significantly greater activation of the lumbar multifidus during back extension when the pelvis is stabilised (*San Juan et al., 2005*), and in addition, muscle activation of the gluteus maximus and biceps femoris is decreased (*Da Silva et al., 2009*). However, contrasting evidence does exist to support trunk extension tasks in producing ILEX fatigue. For example, *Edinborough, Fisher & Steele (2016)* reported that performing kettlebell swings produced transient fatigue in ILEX strength, hypothesising that performing the kettlebell swing exercise as part of a training programme might produce a chronic training effect.

Despite this, a recent study (*Androulakis-Korakakis et al., 2018*) has reported that ILEX strength does not differ between recreationally trained males and both competitive and non-competitive powerlifters (NCPL); a population who regularly train with exercises such as the BBS. Currently reviews of the efficacy of different exercise approaches upon ILEX strength suggest that evidence is limited with respect to many approaches, yet that an ILEX exercise may be most efficacious (*Steele, Bruce-Low & Smith, 2015*). However, devices for this are expensive, not readily available, and as such, the consideration of cheaper and accessible alternatives to ILEX exercise for strengthening the lumbar extensors should be considered.

To date, no empirical studies have assessed the efficacy of either the BBS exercise or the BHT upon ILEX strength. Thus, the aim of the present study was to consider the efficacy of a 4-week mesocycle of either BBS or BHT exercise in increasing ILEX strength.

## METHODS
A randomised trial, research design was used whereby 14 trained males were randomised in to either BBS ($n = 7$) or BHT ($n = 7$) training. Both groups were assessed pre- and post-intervention for BBS and BHT maximal strength (1-repetition maximum; 1RM) as well as isometric ILEX strength. The study was approved by Solent University Health, Exercise, and Sport Science (HESS) ethics committee (ID No. 669).

Using convenience and snowball sampling methods, 14 trained males were recruited. Subjects were required to have >6 months resistance training (RT) experience, and currently be performing a structured RT programme with at least one session per week including the use of BBS exercise and BHT exercise and have no history of low-back pain. Written informed consent was obtained from all subjects prior to participation. Subjects were randomised using a computer randomisation programme to one of two groups; BBS ($n = 7$), or BHT ($n = 7$). Subjects were asked to refrain from any exercise away from the supervised sessions. Participant demographics are included in Table 1.

**Table 1 Participant characteristics.**

| Characteristic | Back squat ($n = 7$) | Barbell hip thrust ($n = 7$) |
| --- | --- | --- |
| Age (years) | 21.71 ± 0.45 | 22.43 ± 0.49 |
| Height (cm) | 181.61 ± 4.63 | 177 ± 7.61 |
| Body mass (kg) | 76.49 ± 9.09 | 83.06 ± 15.81 |

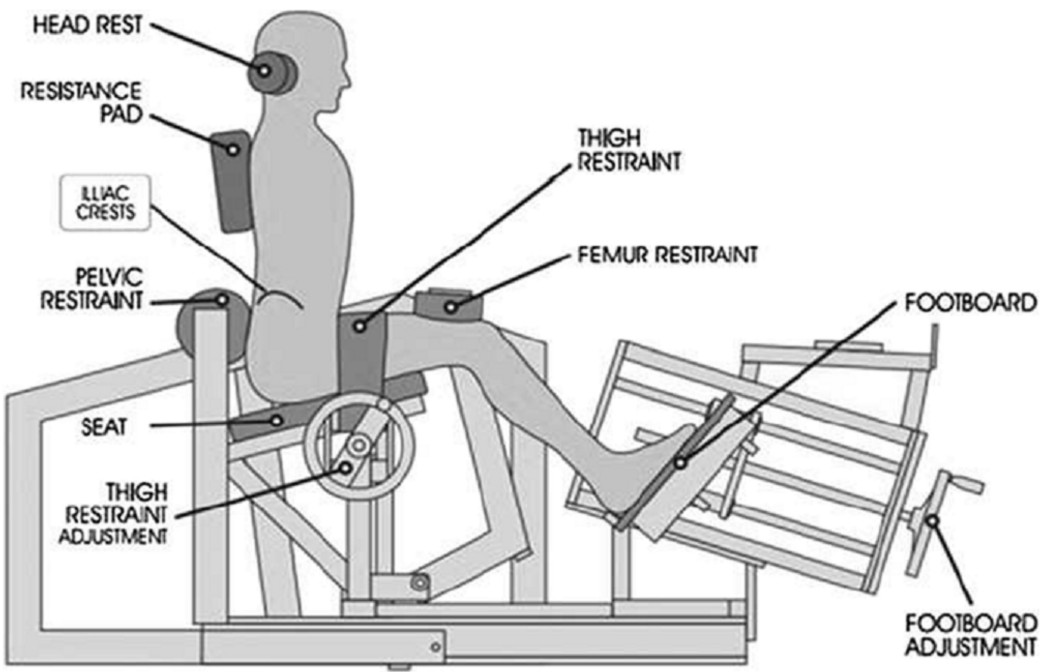

**Figure 1 MedX lumbar extension machine showing restraint system.**

A power analysis of previous research, using similar study design and asymptomatic subjects (*Contreras et al., 2017*; *Styles, Matthews & Comfort, 2016*) was conducted to determine sample size ($n$). The ES was calculated for both studies using *Cohen's d (1992)*, producing a mean ES of 1.67 for BBS and BHT strength increases. Participant numbers were calculated using equations from *Whitley & Ball (2002)* revealing each group required seven subjects to meet required β power of 0.8 at an α value of $p < 0.05$. It should be noted that the study was powered for the identification of within-group changes (two tailed) in outcomes and between group comparisons were a secondary outcome. Based upon sensitivity analysis for between group comparisons for the analysis detailed below the study was powered to identify at most a large between group ES of $f = 0.82$.

Subjects attended a preliminary session where they were assessed on their familiarity with both BBS and BHT exercises and verified their ability to perform them safely with correct technique. They also attended a familiarisation session for the MedX lumbar extension machine (MedX, Ocala, FL, USA) used for measuring isometric ILEX, where subjects performed a testing session in the format detailed below. The ILEX machine (Fig. 1) has been demonstrated as valid (*Graves et al., 1990*, *1994*) and reliable
($r$ = 0.94–0.98; *Pollock et al., 1989*) and the details of which have been described elsewhere (*Graves et al., 1990*).

Subjects reported to the laboratory having refrained from exercise other than that of daily living for at least 48 h before baseline testing, and at least 48 h before post-intervention testing. Maximum strength testing was consistent with recognised guidelines established by the NSCA (*Baechle & Earle, 2008*). Prior to testing, subjects performed a general warm-up consisting of 5 min cycling at 60–70 rpm and 50 w. Next, a specific warm-up set of the prescribed exercise for five repetitions was performed at ~50% 1RM followed by one to two sets of two–three repetitions at a load corresponding to ~60–80% 1RM. Subjects then performed one repetition sets of increasing weight for 1RM determination. The external load was adjusted by ~5–10% in subsequent attempts until the subject was unable to complete one maximal muscle action. The 1RM was considered the highest external load lifted. A 3- to 5-min rest was provided between each attempt. All 1RM determinations were made within five attempts. Test–retest reliability was not determined within the present study, but instead was calculated from previous studies as the typical error of measurement and 95% confidence intervals (CI). For the BBS, this was calculated as 3.8 kg (95% CI = 2.9–5.5 kg) from the average of typical errors of measurement (using average of participant sample sizes to determine degrees of freedom for calculation of 95% CI) reported in studies reviewed on BBS 1RM reliability by *Nuzzo, Taylor & Gandevia (2019)* with a range of 2–4 days between test–retest session. For the BHT, this was calculated as 8.3 kg (95% CI = 5.7–15.2 kg) from the standard deviation of changes from the control group reported by *Jarvis et al. (2019)* over a period of 8 weeks.

The BBS 1RM was completed first, followed by the BHT 1RM, with a 20-min rest interval between exercises to allow for sufficient recovery. As per *Contreras et al. (2015)*, the BHT was performed by having subjects' upper back on a bench with the feet positioned wider than shoulder width and toes pointed forwards or slightly outwards. The barbell was padded with a thick bar pad and placed over the subjects' hips. Subjects were instructed to thrust the bar upwards while maintaining a neutral spine and pelvis. Full extension of the hips (180°) was required for a successful lift.

On a separate day following no less than 72 h rest, subjects attended the laboratory for ILEX strength testing. Subjects were seated upright with their pelvis secured by a restraint pad across the anterior, upper thigh, and another across the thigh just superior to the knee. These pads were fixed tightly to ensure the effort produced was from the lumbar musculature only and not from the pelvis or thighs, isolating the lumbar extensors. A counter-weight was used to balance the mass of the upper body and the effects of gravity on the upper body. All subjects were assessed for range of motion (ROM) and performed a dynamic warm-up with a load equating to 90 lbs/~41 kg and three submaximal isometric tests at full flexion, full extension and a mid-range position. Maximal isometric testing was then performed at seven joint angles (0°, 12°, 24°, 36°, 48°, 60°, and 72° of extension) where subjects were encouraged to gradually achieve maximal effort over 2–3 s and to maintain the maximal contraction for a further 1 s. The torque produced was measured by a load cell attached to the movement arm. Subjects rested for 5–10 s between

tests at different joint angles. Isolated lumbar extension strength was considered as a 'strength index' (SI) calculated as the area under the torque curve (using the trapezoid formula) from multiple angle testing in order to provide a composite measure of overall changes in strength across the ROM (see Supplementary File). Based upon between day repeated test–retest data from prior studies in our lab (*Edinborough, Fisher & Steele, 2016*; *Stuart et al., 2018*) we determined the typical error of measurement for the SI as 1659.51 Nm·degrees (95% CI = 1316.46–2317.17 Nm·degrees) using *Hopkins (2015)* spreadsheets for reliability.

Both the BBS and BHT participant groups attended two RT sessions per week, for a 4-week mesocycle. During these sessions each participant performed three sets using 80% of their 1RM (mean = 8RM; range = 6–10RM) in a controlled, non-explosive manner (2 s concentric, 4 s eccentric; monitored by a supervisor using a metronome) in order to maximise muscle tension and eliminate momentum. All sets were performed to momentary failure (e.g. the inability to complete the concentric phase of a repetition despite maximal effort; *Steele et al., 2017*) whereby if participants completed a repetition then they attempted the successive repetition until they could not complete the concentric phase of the movement. Subjects were asked to confirm maximal effort using a CR-10 rating of perceived exertion scale (*Day et al., 2004*), and were instructed to rest 3–5 min between sets. All training sessions were supervised one-to-one and attempts were made to increase the load lifted each week whilst maintaining the target repetition range. No injuries were reported and adherence to the programme was 100% for both groups.

The independent variable was the group (BBS or BHT) and dependent variables changes (i.e. post-test minus pre-test values) in BBS 1RM, BHT 1RM, and SI. Analysis of covariance (ANCOVA) was used for between group comparisons in dependent variables with baseline measures as covariates (i.e., pre-intervention BBS 1RM, BHT 1RM, and SI). Point estimates were calculated along with the precision of those estimates using 95% CI for within-group adjusted means. The 95% CI were further interpreted to indicate that significant within-group changes occurred if the upper or lower limits do not cross zero. The software used was SPSS (version 23, IBM Corp, Portsmouth, UK) and the cut off for significance was $p < 0.05$. Gardner–Altman plots were also produced using Estimation Statistics (*Claridge-Chang & Assam, 2016*) for data visualisation. Visual inspection of the data using boxplots revealed two outliers (determined using the interquartile rule) for both change in BBS 1RM (in the BHT group) and change in BHT 1RM (in the BBS group) and further Shapiro–Wilk tests confirmed that data did not meet assumptions of normality of distribution for the groups containing the outliers (change in squat 1RM, BHT group $p < 0.001$; change in BHT 1RM, BBS group $p < 0.001$). Thus, for these variables, due to the significant deviations from normality of distribution combined with the relatively small sample size, the data were rank transformed prior to performing ANCOVA (*Olejnik & Algina, 1984*). All results are reported in the units of measurement for each test.

## RESULTS

The 95% CI for changes did not cross zero for change in squat 1RM or change in BHT 1RM in either group and thus both groups had significant within-group improvements

**Table 2 Pre- and post-intervention 1-repetition maximum (1RM; means ±SD) and isolated lumbar extension strength (N.m), changes, and 95% confidence intervals (CI).** Data for back squat 1RM and hip thrust 1RM are presented as medians (±IQR), whereas data for isolated lumbar extension strength is presented as mean (±SD).

| Group | Variable | Pre-intervention | Post-intervention | Changes | 95% CI for change |
|---|---|---|---|---|---|
| Back squat group ($n = 7$) | Back squat 1RM (kg) | 115.0 ± 35.0 | 122.5 ± 27.5 | 10.0 ± 7.5 | [7.6–15.9] |
| | Hip thrust 1RM (kg) | 160.0 ± 42.5 | 180.0 ± 50.0 | 15.0 ± 5.0 | [6.1–34.6] |
| | Isolated lumbar extension strength (Nm·degrees) | 19,670 ± 1,974 | 20,000 ± 1847 | 321 ± 270 | [25–600] |
| Hip thrust group ($n = 7$) | Back squat 1RM (kg) | 110.0 ± 7.5 | 117.5 ± 17.5 | 5 ± 2.5 | [1.8–15.4] |
| | Hip thrust 1RM (kg) | 162.5 ± 25.0 | 187.5 ± 42.5 | 27.5 ± 15.0 | [19.9–35.1] |
| | Isolated lumbar extension strength (Nm·degrees) | 21,310 ± 2,950 | 21,820 ± 2998 | 509 ± 439 | [231–856] |

in these outcomes which also exceeded the typical errors of measure. The 95% CI for changes in SI did not cross zero for the BHT group suggesting a significant change; however, change in SI did not exceed typical error of measurement for either group and so is unlikely to be a meaningful change. Between group comparisons using ANCOVA revealed significant differences for change in squat 1RM ($F_{(1,11)} = 5.240$, $p = 0.043$), change in BHT 1RM ($F_{(1,11)} = 6.673$, $p = 0.025$), but not for change in SI ($F_{(1,11)} = 1.541$, $p = 0.240$). Table 2 shows pre- and post-intervention results, unadjusted means (±SD) or medians (±IQR) for rank transformed variables for changes in each outcome measure, and 95% CI for the changes. For visual depiction of results, Gardner–Altman plots of changes for each outcome are presented in Fig. 2.

## DISCUSSION

To the authors knowledge this is the first study to investigate the effects of BBS and BHT strength training upon ILEX strength. As such, this research adds to a dearth of literature considering exercise protocols to improve lumbar extension strength.

We should first consider the efficacy of the training routine for the specific exercises. Our analyses revealed that the BBS training group significantly improved BBS and BHT 1RM by averages of ~11.8 kg/10.2%, and ~20.3 kg/13.3%, respectively. Furthermore, the BHT training group significantly improved BBS and BHT 1RM by averages of ~8.6 kg/7.7% and ~27.5 kg/24.8%, respectively. Further, these changes exceeded the typical errors of measurement. Whilst specificity meant that both groups improved their strength to a significantly greater degree on the exercise they used during training (e.g. the BBS group improved their BBS 1RM to a greater degree than the BHT group, and the BHT group improved their BHT 1RM to a greater degree than the BBS group), the present data suggest a degree of transferability between exercises. In considering transferability, it is noteworthy that the largest individual strength increase in BBS 1RM (25 kg) was a participant in the BHT group, and the largest individual increase in BHT 1RM (55 kg) was a participant in the BBS group. Whilst our data cannot explain individual differences based on the possible heterogeneity of the participant group, we might reflect on the degree of training status of these participants that accommodates such large strength increases

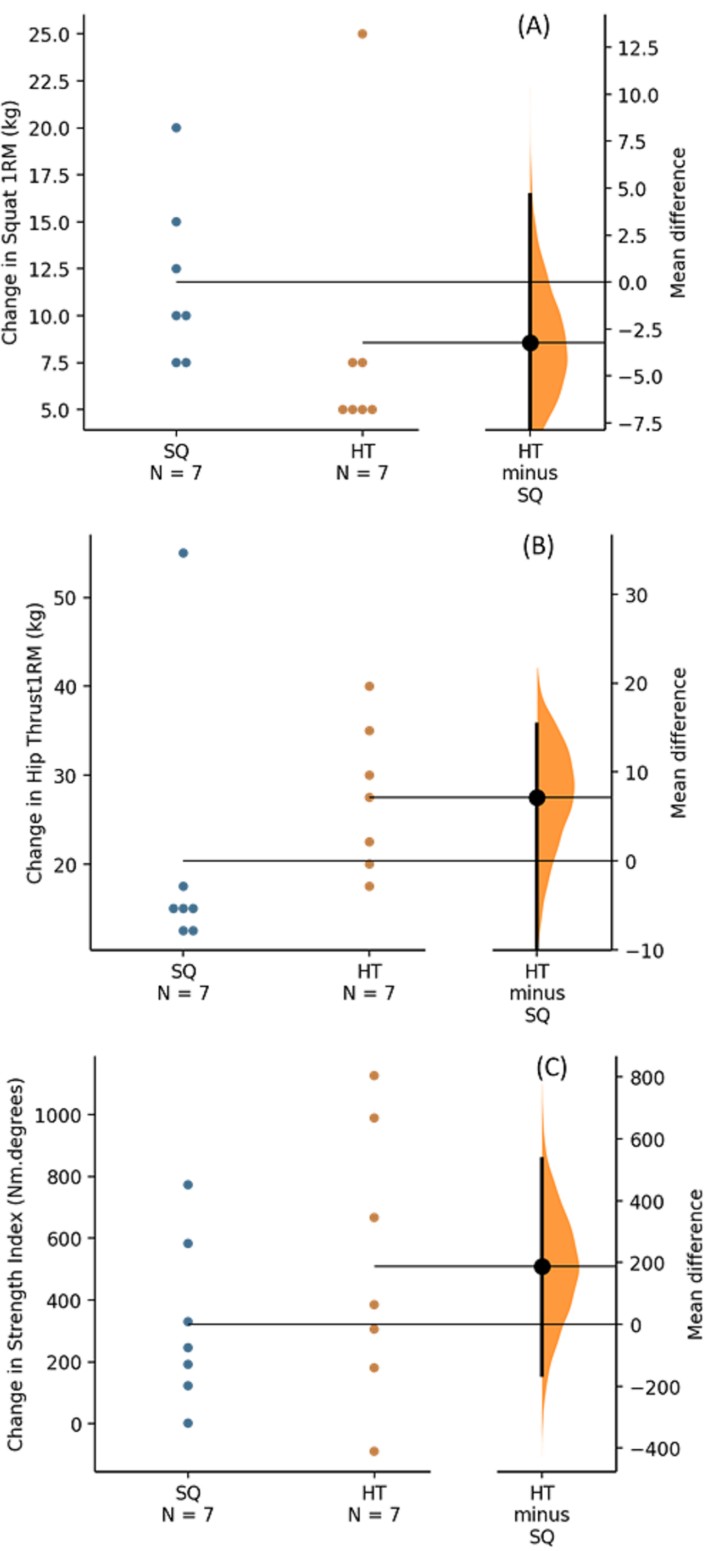

**Figure 2 Gardner–Altman plots – all participants.** Figure shows change for each outcome for all participants ((A) Change in squat 1RM, (B) Change in hip thrust 1RM, (C) Change in ILEX strength index).

over a 4-week training cycle. Any attempt at explaining these large individual changes would be purely speculative; however, we postulate that they might have arisen from the mesocycle prior to this intervention period not being focused on maximal strength, and/or that participants might have benefited from direct individual supervision, as has been seen in previous research with some training experience (*Coutts, Murphy & Dascombe, 2004*). As such the participants in both the BBS and BHT group might never have trained to the same *intensity of effort* and as such made significant strength increases which showed transferability. In addition the participant in the BHT group might have benefited from practice of recruiting the gluteal muscles which, in turn, supported improvement in the BBS 1RM. For example previous research by *Crow et al. (2012)* supports acute improvements in counter-movement jump following specific gluteal exercises. It might be that there are chronic adaptations to multi-joint lower-body movements that support the targeting of the gluteal muscle group as part of a training intervention.

However, our data suggest that neither the BBS nor BHT RT exercises serve to improve ILEX strength, since the pre- to post-intervention changes did not exceed the typical error of measurement. Recent research serves to support our findings in context of the BBS exercise. For example, *Vigotsky et al. (2019)*, recently reported no relationship between BBS 1RM and isometric spinal extension strength. In their discussion, Vigotsky et al. suggested that, during a BBS exercise, 'the spinal erectors need only resist the net joint moment as well as a small abdominal co-contraction, which does not necessarily increase with load'. Additionally, *Androulakis-Korakakis et al. (2018)* recently reported isometric lumbar extension torque and BBS 1RM values for NCPL and competitive powerlifters (CPL). The data suggested that, despite large and significant differences in BBS 1RM (NCPL = 177.0 kg, CPL = 215.2 kg), there were no differences in ILEX strength (SI; NCPL = 22,864 Nm·degrees, CPL = 22,850 Nm·degrees). Indeed, data from the present study produced similar ILEX strength values 20,000–21,820 Nm·degrees supporting that beyond a certain threshold the lumbar extensors might not be required to increase in strength to aid BBS performance. However, despite the lack of association between ILEX strength and BBS strength, and the existence of an association between performance markers and the BBS, it is not wholly clear whether *increasing* lumbar extension strength through isolated training might also increase performance. The absence of specific isolated lumbar extensor training may be suboptimal for developing athletic performance. Indeed, *Fisher, Bruce-low & Smith (2013)* have shown that ILEX RT can increase Romanian deadlift 1RM. Further research is required though to examine ILEX training interventions upon performance outcomes, including BBS strength and sporting performance.

The BHT represents a more contemporary, and thus limited, area of exercise science research. Certainly evidence has suggested that the BHT might be an efficacious exercise for improving sprint performance and mid-thigh pull (*Contreras et al., 2017*), and data support considerable muscle activation of the hamstring and gluteal muscles (*Contreras et al., 2015*). In addition, research has suggested greater erector spinae muscle activation for the BHT compared to the barbell- and hex bar-deadlifts (*Andersen et al., 2018*), and as noted improving ILEX strength can serve to improve Romanian deadlift 1RM (*Fisher, Bruce-low & Smith, 2013*). However, the present study suggests that training,

and indeed enhancing 1RM for the BHT exercise does not improve ILEX strength. This specificity of adaptation is further supported as *Fisher, Bruce-low & Smith (2013)* also reported a group training using the Romanian deadlift increased their Romanian deadlift 1RM significantly but failed to increase their ILEX strength.

With the above in mind it appears that, despite large increases in BBS and BHT 1RM as a result of the respective RT programmes, ILEX strength is likely not improved by either exercise. Previous research has suggested that the use of exercise where pelvic rotation is permitted does not improve ILEX strength (*Graves et al., 1994*; *Fisher, Bruce-low & Smith, 2013*). In context, it might be that the gluteal and hamstring muscles serve to provide trunk extension (e.g. hip- and lumbar extension), and as a result both BBS and BHT RT produce strength increases in these muscles which can transfer to improve performance between the respective exercises. However, as a result of the pelvic rotation through trunk extension (and thus the dominance of the gluteal and hamstring muscles), neither BBS nor BHT exercises appear to provide a sufficient training stimulus to the lumbar extensors. This is fitting with previous research which has reported no relationship between trunk extension performance (assessed via Biering-Sorensen test) and ILEX strength in both asymptomatic persons as well as those symptomatic with chronic low-back pain (*Conway et al., 2017*). Nonetheless, some tasks performed with pelvic rotation can induce lumbar fatigue suggesting a role for this muscle during movement, for example kettlebell swings (*Edinborough, Fisher & Steele, 2016*). Though the present data suggest that both the BBS and BHT do not improve lumbar extension strength, and previous data suggest the Romanian deadlift also lacks efficacy (*Fisher, Bruce-low & Smith, 2013*), further research should consider other exercises such as kettlebell swings in training interventions.

We should, of course, remember that exercises such as the BBS are not performed solely with the intent of strengthening the lumbar extensors, and that this exercise shows a strong relationship to athletic performance markers such as sprint speed ($r = 0.71$–$0.94$) and vertical jump ($r = 0.78$; *Wisløff et al., 2004*). However, a previous review has questioned the need for adding single-joint exercise to a RT programme since muscular adaptations appear similar to when performing only multi-joint exercises (*Gentil, Fisher & Steele, 2017*). As mentioned above we might consider trunk extension (e.g. hip and lumbar extension) to be multi-joint, and ILEX to be more similar to single-joint movements (though strictly speaking it is a multi-joint movement due to the vertebral segments). However, in light of the previous research as well as present findings, it appears that multi-joint exercises which include trunk extension (such as the BBS and BHT) are not sufficient to strengthen the lumbar extensors. As such, though multi-joint movements may be sufficient for appendicular muscular adaptations (*Gentil, Fisher & Steele, 2017*), we would suggest that both athletes and lay persons consider supplementing existing training practices with specific ILEX exercise to strengthen the lumbar muscles.

Whilst the present study provides useful data and guidance as to the efficacy of different exercises in strengthening the lumbar extensors, we should accept the limitation that we did not include an ILEX training group which might reflect the possible comparative increases in ILEX strength. Previous research has demonstrated that isometric ILEX

strength can increase considerably as a result of once weekly training sessions using 80% MVC over 10-weeks (SI change = 4,353 Nm·degrees; *Fisher, Bruce-low & Smith, 2013*). Furthermore, interventions performed once a week over 6 week durations have been shown to produce significant increases in lumbar extension strength using both single- and multiple-sets in trained males (single set = 1,854 Nm·degrees, multiple set = 2,415 Nm·degrees; *Steele et al., 2015*), and with both heavier-loads (80% MVC) and lighter-loads (50% MVC) in recreationally active males and females (SI change; 80% MVC = 2,891 Nm·degrees, 50% MVC = 2,865 Nm·degrees; *Fisher et al., 2018*). As such, had we included an ILEX training group, we would expect eight sessions performed over 4 weeks to have likely produced significant and meaningful increases in isometric ILEX strength. A further limitation might be the brevity of the present 4-week strength mesocycle. Whilst we contest that this is fitting with training practices, we accept that many athletes and persons undertake longer mesocycles, or continue exercises across multiple phases of periodisation. Future research might consider the transference of adaptations as a result of continued BBS or BHT RT through strength, power and/or hypertrophy loading phases. Lastly, this study was relatively small and primarily powered to identify within-group changes in outcomes. Though we did identify between group differences in both BBS and BHT 1RM changes future research with greater sample sizes should examine this with greater statistical power. Finally, our analysis considered lumbar strength increases across the entire range of movement (SI, calculated as the area under the strength curve). However, it might be that analyses of individual angles would reveal angle specific strength increases—particularly at 0° (full extension) where the hip thrust exercise might have provided the greatest activation of the lumbar extensors (e.g. at 180° during a BHT exercise; *Andersen et al., 2018*). Future research might consider sufficient sample sizes and power to examine this as a pre-specified hypothesis.

## CONCLUSION

The present study provides support for the concept of specificity, particularly in relation to the movement mechanics between trunk extension (including pelvic rotation) and ILEX. Our data suggest that both the BBS and BHT exercises produce meaningful increases in strength which might transfer between lower-body trunk extension exercises.
This allows strength coaches, personal trainers, and trainees to self-select multi-joint lower-body trunk extension exercises based on preference or variety. However, evidence suggests that neither the BBS nor BHT exercises, nor indeed any exercise allowing pelvic rotation through trunk extension, can meaningfully increase ILEX strength. Since strengthening these muscles might enhance physical and sporting performance we encourage strength coaches and personal trainers to supplement existing practices by prescribing specific ILEX exercise.

### Funding
The authors received no funding for this work.

## Competing Interests

The authors declare that they have no competing interests.

## Author Contributions

- Alexander Hammond performed the experiments, analysed the data, contributed reagents/materials/analysis tools, prepared figures and/or tables, authored or reviewed drafts of the paper, approved the final draft.
- Craig Perrin performed the experiments, analysed the data, contributed reagents/materials/analysis tools, prepared figures and/or tables, authored or reviewed drafts of the paper, approved the final draft.
- James Steele performed the experiments, analysed the data, contributed reagents/materials/analysis tools, prepared figures and/or tables, authored or reviewed drafts of the paper, approved the final draft.
- Jürgen Giessing analysed the data, contributed reagents/materials/analysis tools, prepared figures and/or tables, authored or reviewed drafts of the paper, approved the final draft.
- Paulo Gentil analysed the data, contributed reagents/materials/analysis tools, prepared figures and/or tables, authored or reviewed drafts of the paper, approved the final draft.
- James P. Fisher conceived and designed the experiments, performed the experiments, analysed the data, contributed reagents/materials/analysis tools, prepared figures and/or tables, authored or reviewed drafts of the paper, approved the final draft.

## Human Ethics

The following information was supplied relating to ethical approvals (i.e., approving body and any reference numbers):

Solent University Health, Exercise and Sport Science (HESS) ethics committee approved this study (ID No. 669).

## Data Availability

Raw data are available as a Supplemental File.

## Supplemental Information

Supplemental information for this article can be found online at http://dx.doi.org/10.7717/peerj.7337#supplemental-information.

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
