# Peer review of "The effects of a 4-week mesocycle of barbell back squat or barbell hip thrust strength training upon isolated lumbar extension strength"

_PeerJ, doi:10.7717/peerj.7337_

## Round 0.1 · original submission · Major Revisions

Based on the comments from both reviewers I am happy to recommend you be allowed to respond to the reviewers comments. Please pay particular attention to reviewer twos comments.

·

Basic reporting

No comment

Experimental design

No comment

Validity of the findings

No comment

Additional comments

This is a well-written paper. I especially enjoyed the intro and discussion sections. I would like to see this paper get published as it shows that hip extension exercises, contrary to popular opinion in S&C, do not sufficiently strengthen the erectors. I have just a few minor points for the authors to address prior to accepting the article for publication:

Abstract page 6 line 47: put comma after strength coaches

You may want to point this out as I found it to be intriguing. The individual who gained the most squat strength (25 kg) was someone from the hip thrust group, and the individual who gained the most hip thrust strength (55 kg) was someone from the squat group. Gaining 25 kg on the squat or 55 kg on the hip thrust in 8 sessions over 4 weeks for someone with at least 6 months of training experience is very impressive.

Last thing. I wonder if any particular joint angle of lumbar extension increased significantly. I understand that the area under the curve (strength index) didn't, but maybe the measurement for neutral (0 degrees) did. Did you guys look into this at all? If so, please address it. If not, please list it as a limitation. I appreciate you mentioning the brevity and small sample sizes in the discussion.

Reviewer 2 ·

Basic reporting

The authors completed a very small study powered to look at within group changes to two different forms of exercise. The article is very well written and the authors are very clear that they are underpowered to look at between group differences. The authors demonstrate the principle of specificity but also suggest that there may be some degree of cross-over to other strength tasks. Below, I offer suggestions that may be able to improve the manuscript

Introduction line 58: I wonder if it makes sense to say that 50% of the variance is shared rather than “it explains 50% of the variance.” Perhaps this is semantics but sometimes this leads people to think in a “causal” way.

Methods: For your analysis, I think it would be quite useful if you were able to provide time matched control assessments for your 1RM and hip thrust strength tasks. For example, even using an estimate may be better than nothing. For example centering it around 0 and then add in what you might expect that SD of the difference to be…that would allow you to calculate if your within group comparisons are different than noise. I think this would be much stronger than just stating that it is different than 0 (pre to post difference). It would be much stronger if you could build in your “reliability” to an actual statistical comparison.

You could do the same thing with your SI index. I was a bit confused on the reliability that you propose in the text because it is in Nm/degrees whereas data is represented as Nm only.

Results line 211: I agree that nothing is going on here, or very little, but I do not think it is necessarily because the mean does not exceed the minimal difference. Typically that is used to determine the confidence of one person exceeding this threshold but I do not think the entire mean change has to surpass the “minimal difference”. I think you could do like the earlier suggestion, and just compare the effects of the treatment to that of the “pseudo control”. Same comment at line 232.

Figures: if you could add in a line to denote “zero”…that might be easier for readers to orient themselves to the figure.

Experimental design

see above

Validity of the findings

see above

Additional comments

see above

---

## Round 0.2 · Minor Revisions

I thank the authors for their hard work intending to the reviewer's comments on the initial version of this manuscript. The paper would be accepted for publication with the very minor exception regarding the strength index. At this stage, on line 180 the definition provided is a little bit unclear. Can this please be clarified, perhaps with a representative picture of the torque curve? Further, you haven't attended to the second reviewer's comments regarding the units of measurement for the strength index. On line 184, you state that the units of measurement are Nm.degrees but throughout the rest of the manuscript report the units as just Nm. Once these issues are clarified, I would be happy to recommend this paper be accepted for publication.

·

Basic reporting

All good

Experimental design

All good

Validity of the findings

All good

Additional comments

Great job with the edits and explanations.

Reviewer 2 ·

Basic reporting

see below

Experimental design

see below

Validity of the findings

see below

Additional comments

I appreciate the authors addressing my questions. My only remaining comment is that SI index is still Nm in the figure rather than Nm-deg and same comment in Table 2.

---

## Round 0.3 · accepted · Accept

We thank the authors for their amendments and am happy to recommend this manuscript be accepted for publication.